# Bayesian Q-learning With Imperfect Expert Demonstrations

**Fengdi Che**[*]
University of Alberta

**Xiru Zhu** [†]
McGill University

**Doina Precup**
McGill University, MILA, DeepMind

**David Meger**
McGill University

**Gregory Dudek**
McGill University, Samsung

## Abstract

Guided exploration with expert demonstrations improves data efficiency for reinforcement learning, but current algorithms often overuse expert information. We propose a novel algorithm to speed up Q-learning with the help of a limited amount of imperfect expert demonstrations. The algorithm avoids excessive reliance on expert data by relaxing the optimal expert assumption and gradually reducing the usage of uninformative expert data according to the uncertainty of learning results. Experimentally, we evaluate our approach on a sparse-reward chain environment and six more complicated Atari games with delayed rewards. With the proposed methods, we can achieve better results than Deep Q-learning from Demonstrations (Hester et al., 2017) in most environments.

## 1  Introduction

Reinforcement learning (RL) trains an agent to maximize expected cumulative rewards through online interactions with an environment [Sutton and Barto, 2018]. To speed up learning, online RL can be combined with offline expert demonstrations [Hester et al., 2018], which guides the agent toward high-rewarding behaviours and thus improves data efficiency. Most existing expert demonstration methods [Brys et al., 2015, Abbeel and Ng, 2004] either clone behaviours from the expert or bonus expert actions. However, agents are provided with imperfect expert data and gain distracting guidance. Some works start handling the imperfect expert data but still lack solid theoretical foundation [Nair et al., 2018, Zhang et al., 2022].

Our paper adopts Bayesian frameworks [Ghavamzadeh et al., 2015, Dearden et al., 1998], which provide an analytical method to incorporate extra sub-optimal expert information for learning. In the probabilistic model for Bayesian inference, the sub-optimal expert decision is assumed to rely on the Boltzmann distribution dependent on the optimal expected returns from state-action pairs, called the optimal Q-values. Intuitively, our paper considers that the expert would prefer actions with higher optimal Q-values. Based on this relationship, the agent can infer values of the optimal Q-values from the given expert data during online learning and correct estimated Q-values to better correspond to the expert behaviours. This inference is equivalent to maximizing the posterior distribution of Q-values conditioned on expert data, but computing the maximum of a posterior probability is a difficult task [Hoffman et al., 2013, Diaconis and Ylvisaker, 1979, West et al., 1985]. Therefore, our paper proposes to utilize the generalized extended Kalman filter (GEKF) [Fahrmeir, 1992] to derive a posterior maximum, requiring fewer restrictions on Q-values functions than other Bayesian RL frameworks [Dearden et al., 1998, Engel et al., 2003, Osband et al., 2019].

---

[*]equal contribution
[†]equal contribution

Offline Reinforcement Learning Workshop at Neural Information Processing Systems, 2022

Under the assumptions from the GEKF framework, we derive an iterative forward way to compute Q-values, which consists of a Q-value-update step as in the Q-learning algorithm and an expert correction step. In the correction step, our update rule weighs expert information according to an agent's uncertainty of its self-learning result, measured by the estimated posterior variance of learned Q-values. Thus, the larger Q-values' variance is, the more our expert correction encourages expert behaviours by increasing expert actions' Q-values and decreasing non-experts' Q-values. This mechanism reduces the influence of uninformative expert data and avoids excessive guided exploration.

Furthermore, we propose our computationally efficient deep algorithm, Bayesian Q-learning from Demonstrations (BQfD), built on top of the Deep Q-learning Network (DQN) [Mnih et al., 2013]. The algorithm embeds reliable expert knowledge into Q-values and leads to a more efficient exploration, as shown on a sparse-reward chain environment DeepSea and six randomly chosen Atari games. In most environments, our algorithm learns faster than Deep Q-learning from Demonstrations (DQfD) [Hester et al., 2018] and Prioritized Double Duelling (PDD) DQN [Wang et al., 2016].

## 2 Related Work

Reinforcement learning from demonstrations (RLfD) has drawn attention in recent years. This method only requires a small number of offline expert demonstrations and can noticeably improve performance. Deep Q-learning from demonstrations (DQfD) [Hester et al., 2018] includes an additional margin classification loss to ensure that $Q$-values of expert actions are higher than other actions. Reinforcement learning from demonstrations through shaping [Brys et al., 2015] assigns high potential to a state-action pair $(s, a)$ when the action $a$ is used by the expert in the neighbourhood of the state $s$. Wu et al. [Wu et al., 2020] further leverages reward potential computed by generative models. However, this requires an added complexity of training generative models. Expert data is also used implicitly to learn rewards inversely at the beginning [Abbeel and Ng, 2004, Brown et al., 2019].

However, these approaches do not consider the case where the expert is imperfect and can often have difficulty exceeding expert performance. In contrast, Nair et al. [Nair et al., 2018] handle suboptimal demonstrations by only cloning expert actions when their current estimated Q-values are higher than others. Zhang et al. [Zhang et al., 2022] utilizes expert information when the cumulative Q-values on expert state-action pairs are larger than cumulative value functions. However, the learned Q-value is often unreliable and frequently changes, making it a poor judgment of expert data quality. At the same time, our measurement based on the posterior variance of estimated Q-values is more reasonable.

Jing et al. [Jing et al., 2020] models the suboptimal expert policy as a local optimum to maximize the expected returns and then constrains the agent to learn within a region around the expert policy, measured by KL divergence between occupancy measures. However, the local optimum condition is hard to satisfy, and the limitation of learning around the expert policy cannot deal with mistaken expert actions. In contrast, our assumption on Boltzmann distributed expert policy is much weaker.

## 3 Background

A Markov decision process [Sutton and Barto, 2018] is a tuple $M = \langle \mathcal{S}, \mathcal{A}, P, \gamma, R, \rho \rangle$ where $\mathcal{S}$ is the state space, $\mathcal{A}$ is the action space, $P$ is the time-homogeneous transition probability matrix with $P(s'|s, a)$ as elements, $\gamma \in (0, 1)$ is the discount factor, $R$ is the reward function, with $R(s, a)$ denoting a random vector describing rewards received after state-action pair $(s, a)$ and $\rho$ is the distribution of the initial state. At each time step $h$, the agent samples an action $A_h$ from a policy $\mu$, and then transits to the next state $S_{h+1}$ and gains a reward $R(S_h, A_h)$. Our paper focuses on finite horizon cases with horizon $H$, where an agent stops at time step $H$. Also, our paper works on finite action and state spaces.

The expected discounted cumulative return starting from each state-action pair $(s, a)$ at time step $h$ and following the policy $\mu$ is called the Q-value, denoted by $q_h^\mu(s, a)$, and is defined as follows:

$$q_h^\mu(s, a) = \mathbb{E}_\tau[\sum_{t=h}^{H} \gamma^t R(S_t, A_t)],$$

where $\tau$ denotes trajectories with $S_h = s$, $A_h = a$, $S_t \sim P(\cdot|S_{t-1}, A_{t-1})$, $A_t \sim \mu(\cdot|S_h)$ and rewards $R(S_t, A_t)$. The unique optimal Q-value function at time $h$ is defined as:

$$q_h^*(s, a) = q_h^{\mu^*}(s, a) = \sup_{\mu} q_h^{\mu}(s, a).$$

The optimal Q-values also satisfy the optimal Bellman equation for all states and actions:

$$q_h^*(s, a) = \mathbb{E}[R(s, a)] + \gamma \mathbb{E}_{S' \sim P(\cdot|s,a)}[\max_{a'} q_{h+1}^*(S', a')] =: \mathcal{T}_{h+1} q_{h+1}^*(s, a), h = 0, \cdots, H - 1$$

$$q_H^*(s, a) = 0, \ \forall(s, a). \tag{1}$$

The optimal Bellman operator $\mathcal{T}$ is defined by the equation. Moreover, the optimal Q-values at each time $h$ for all state-action pairs can be combined into one vector, $q_h^* \in \mathbb{R}^{|\mathcal{S}| \times |\mathcal{A}|}$, each element representing the optimal Q-value for a state-action pair.

### Q-learning

Optimal Q-values can be computed by Q-learning [Watkins and Dayan, 1992, Jin et al., 2018], which estimates Q-values directly and is based on the following update rule:

$$Q_h(s, a) = (1 - \alpha)Q_h(s, a) + \alpha[R(s, a) + \gamma \max_{a'} Q_{h+1}(S', a'),$$

where $Q$ is an estimation of Q-values, $S' \sim P(\cdot|s, a)$ is a sampled next state, and $\alpha \in (0, 1)$ is the learning rate. A choice of adaptive learning rate is to assign each state-action pair a learning rate and decay it with respect to the number of visitations to the state-action pair. We use $n^l(s, a)$ to represent the number of visitation times of state-action pair $(s, a)$ until and including episode $l$.

### Bayesian Model-free Framework

Bayesian reinforcement learning assumes that there is an initial guess or a prior probability over the model parameters, denoted by $P(\mathcal{M})$. Then in the model-based case, the agent gradually learns a posterior probability for the model parameters conditioned on observed trajectories [Duff, 2003]. Next, an agent can learn a policy based on the most likely MDP or by sampling MDPs from the posterior. In the model-free case, we treat Q-values as random variables and implicitly embed parametric uncertainties from different MDPs. A prior distribution of the optimal Q-values at time $h$ is defined as:

$$P(q_h^*(s, a) \leq c) = P(\{\mathcal{M} : q_{\mathcal{M},h}^*(s, a) \leq c\}).$$

Then algorithms compute the posterior probability of Q-values [Dearden et al., 1998, Osband et al., 2018].

## 4 Model

In this section, we present the probabilistic model used for Bayesian inference and then derive the update rule of Q-values by maximizing posterior probability.

### 4.1 Probabilistic Model of Suboptimal Expert Actions

We start by modeling the relationship between expert actions and the optimal Q-values. The expert chooses the optimal actions most frequently but may contain mistakes and select low probability actions. The Boltzmann distribution can describe this behavior with the optimal Q-values as parameters. Under this distribution, expert actions are sampled proportional to the exponential of optimal Q-values up to a constant multiplier. As known, this assumed expert policy maximizes the expected returns regularized by the entropy [Haarnoja et al., 2018], describing almost optimal but sometimes mistaken expert behaviors. The above assumption is formally presented as follows and shown on the left of figure 1, where the expert action at time $h$ and state $s$ is denoted by $A_{exp,h}(s)$.

**Assumption 1.** *Assume that expert demonstrations are drawn from a policy $\pi_{expert}$ dependent on the optimal Q-values. Also, this policy follows the Boltzmann distribution:*

$$\pi_{expert}(a|s) = \frac{e^{\eta q^*(s,a)}}{\sum_{b \in \mathcal{A}} e^{\eta q^*(s,b)}}, \tag{2}$$

*where $\eta$ is any positive constant.*

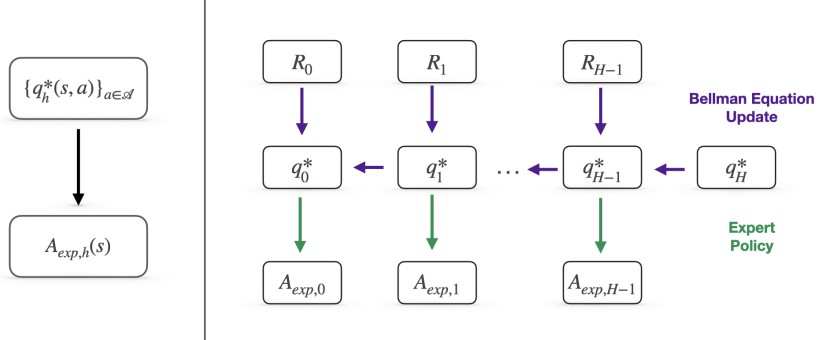

Figure 1: (Left) The figure demonstrates the relationship between optimal Q-values and expert actions. Expert actions are sampled proportional to the exponential of optimal Q-values up to a constant multiplier. (Right) The figure also describes the Bellman update rule for the Q-values along a trajectory.

Moreover, we capture the relationship between the optimal Q-values at different time steps in our probabilistic model through the Bellman equation 1. However, the Bellman equation cannot be computed without the knowledge of environment dynamics. Therefore, our model relies on samples of the reward and the next state, which are widely accepted in the field [Watkins and Dayan, 1992]. We re-write the Bellman equation in a stochastic form as

$$q^*(s, a) = R(s, a) + max_{a' \in \mathcal{A}} \gamma q^*(S', a') + \nu, \tag{3}$$

where $R(s, a)$ and the next state $S'$ are sampled according to the rules of the underlying MDP. The random noise $\nu$ is defined as

$$\nu = \mathbb{E}[R(s, a) + max_{a' \in \mathcal{A}} \gamma q^*(S', a')] - R(s, a) - max_{a' \in \mathcal{A}} \gamma q^*(S', a'),$$

which captures the randomness from the dynamics. Then, the whole model of relationships among rewards, optimal Q-values, and expert actions along a trajectory is presented on the right of figure 1.

## 4.2  GEKF Framework

Next, our desired estimated Q-values should not only follow the stochastic Bellman equation, but they should also most likely give the Boltzmann distribution that the expert is following. This objective is equivalent to maximizing the posterior probabilities of optimal Q-values equaling our estimations conditioned on expert information while constraining to the stochastic Bellman equation.

In order to compute the posterior probability, we need help from a framework under which the posterior probability density function or the posterior mode is of analytical form. Thus, our paper leverages the generalized extended Kalman filter (GEKF) [Fahrmeir, 1992], which can analyze the time series of Q-values with extra expert information and provides an estimation of the posterior maximum. This framework requires extra expert information following an exponential family distribution, which is already assumed.

Furthermore, the framework requires the random noise $\nu$ in the stochastic Bellman equation 3 at each Q-value update step to be Gaussian, which is also assumed in Osband et al. (2019) [Osband et al., 2019]. Meanwhile, this assumption does not hurt in the latter stage of training since the influence of the random noise gradually approaches zero as the number of samples increases, and the learning rate decays. Notice that the random noise should have a zero expectation and bounded variance when rewards and time horizons are bounded. Thus, our paper considers $\nu$ as a Gaussian random variable with zero expectation and a fixed variance $\lambda$, as shown in the following assumption.

**Assumption 2.** *For all state-action pair $(s, a)$, the noise $\nu$ is modeled independently by a Gaussian random variable $\nu \sim N(0, \lambda)$.*

Also, the GEKF framework approximates the posterior distribution by Gaussian and treats the maximization operator in Q-value updates linearly as in Osband et al. (2019) [Osband et al., 2019].

## 4.3 Derivation of Q-values Update Rules

We derive our Q-value estimations $Q$ under the above assumptions given by the GEKF framework [Fahrmeir, 1992]. Firstly, this estimation should satisfy the stochastic Bellman equation in equation 3. Secondly, it should maximize the posterior probability of the optimal Q-values conditioned on expert demonstrations. This task is stated as follows:

$$\max_{Q_{0:H-1}} P(q_{0:H-1}^* = Q_{0:H-1} | A_{exp,0:H-1}, R_{0:H-1}) \tag{4}$$

$$s.t. \quad q_H^*(s,a) = 0, \ \forall s \in \mathcal{S} \text{ and } \forall a \in \mathcal{A}$$

$$q_{h-1}^*(s,a) = R_{h-1}(s,a) + \gamma \max_{a'} q_h^*(S',a') + \nu, \ h = H, \cdots, 1, \ \forall s \in \mathcal{S} \text{ and } \forall a \in \mathcal{A}$$

Recall that $q_h^* \in \mathbb{R}^{|\mathcal{S}||\mathcal{A}|}$ represents the Q-value vector at step $h$ with each element as the value of one state-action pair, and similar for $R_h$; moreover, $A_{exp,h} \in \mathbb{R}^{|\mathcal{S}|}$ represents the expert actions chosen at all states at step $h$. Then, we can eliminate terms independent of Q-values in the posterior and obtain:

$$P(q_{0:H}^* | A_{exp,0:H-1}, R_{0:H-1}) \propto \prod_{h=0}^{H-1} P(A_{exp,h} | q_h^*) \prod_{h=0}^{H-1} P(q_h^* | q_{h+1}^*, R_h) P(q_H^*),$$

where the three distributions are modeled through our previous assumptions: the expert action follows a Boltzmann distribution, Q-values at step $h$ follow a Gaussian distribution with the expectation given by the Bellman equation, and Q-values at step $H$ are all set to be zero.

Next, the task in equation 4 forms a constrained optimization problem and can further be transformed into a two-point boundary value problem [Fahrmeir, 1992, Sage and Melsa, 1971]. This boundary value problem is then solved by the invariant embedding method, which provides approximate solutions for non-linear partial differential equations. The derivation is shown in the appendix following the steps in Sage (1997) [Sage and Melsa, 1971].

Finally, in a backward view, our estimations can be computed step-by-step as presented in algorithm 1. In the prediction step, the algorithm updates Q-values based on sampled rewards and sampled next states by the Bellman equation as the Q-learning algorithm [Watkins and Dayan, 1992]. This update of estimated Q-values can be written as

$$Q_h = R_h + T_h Q_{h+1},$$

using a transformation matrix $T \in \mathbb{R}^{|\mathcal{S}||\mathcal{A}| \times |\mathcal{S}||\mathcal{A}|}$. Its element at each state-action pair and its next-state-maximal-action pair equals to one, that is $T[(s,a),(S',a')] = 1$ where $S' \sim P(\cdot | s,a)$ and $a' = argmax_b Q_{h+1}(S',b)$. And all other elements are zero.

Then, the algorithm computes the estimated posterior variance $W$. Our derivation gives that this variance not only depends on the transformation matrix $T$, but also the Hessian of log expert policy $U \in \mathbb{R}^{|\mathcal{S}||\mathcal{A}| \times |\mathcal{S}||\mathcal{A}|}$.

After computing the variance, expert information corrects Q-values as

$$Q_h(s,a) = Q_h(s,a) + \eta W_h[(s,a),(s,a)][\mathbb{1}[A_{exp,h}(s) = a] - \frac{exp(\eta Q_h(s,a))}{\sum_{b \in \mathcal{A}} exp(\eta Q_h(s,b))}].$$

Recall that $\eta$ is the parameter of the expert policy. So a bonus weighted by the variance is given to expert actions and encourages the agent to explore them when it is not confident of the current learning result. When the variance of a Q-value estimate is low, the agent stops guided exploration by the expert and follows its own learnt policy. Thus, the agent can deal with inadequate expert data by gradually ignoring it. Meanwhile, this expert modification does not break the stochastic Bellman equation in equation 3 since it is listed as the constraint in the posterior maximization task in equation 4.

## 5 Algorithm

Our paper proposes an algorithm Bayesian Q-learning from Demonstrations (BQfD) to realize the GEKF update rules of Q-values as shown in algorithm 1. This algorithm samples a trajectory at

---

**Algorithm 1** GEKF update rule for one episode

---

**Initialize:** Q-values at the last step $Q_H = 0$, Weight matrices of Q-value functions $W_H = \mathbf{0}^{|\mathcal{S}||\mathcal{A}| \times |\mathcal{S}||\mathcal{A}|}$, and Identity matrix $I$.

**for** time step $h = H - 1, ..., 0$ **do**

  **Prediction step:**

  Update Q-value of each state-action pair based on sampled reward and next state though Bellman equation

  $Q_h(s, a) = R_h(s, a) + T_h Q_{h+1}, \forall s, a$

  **Correction step:**

  Compute the estimated posterior variance

  $W_h = T_h^T W_{h+1} T_h + \lambda I$

  $W_h = (W_h^{-1} + U_h)^{-1}$

  Embed in the expert correction

  $Q_h(s, a) += +\eta W_h[(s, a), (s, a)][\mathbb{1}[A_{exp,h}(s) = a] - \frac{exp(\eta Q_h(s,a))}{\sum_{b \in \mathcal{A}} exp(\eta Q_h(s,b))}]$

**end for**

---

the beginning of each turn and then updates Q-values by the sampled-based Bellman rule and the expert correction backward. Thus, our algorithm only updates the Q-value of the state-action pair $(S_h, A_h)$ at each time step $h$ and introduces a learning rate $\alpha$ for Bellman equation updates to reduce variances. This learning rate $\alpha_{n(s,a)}(s, a)$ is considered to be state-action pair dependent and decays as the number of that pair's visitations $n(s, a)$ increases.

However, the posterior variance calculation in the correction step is computationally inefficient due to the Hessian and inverse operations. To avoid these computations, we would like to approximate the posterior variance at a state-action pair $(s, a)$. Given a first-order decaying learning rate $\alpha_{n(s,a)}(s, a) = \frac{1}{\beta + n(s,a)}$, short written as $\alpha$ and ignoring the variance change by expert information, the variance is roughly a function of the visitation times, denoted by $w_{n(s,a)} = O(\frac{1}{(n(s,a))}\lambda)$ and it is similar to count-based exploration [Bellemare et al., 2016].

Furthermore, our paper utilizes deep neural networks and proposes a deep BQfD based on Prioritized Dueling Double (PDD) Deep Q-learning Network (DQN) [Mnih et al., 2013, Wang et al., 2016, Hessel et al., 2018]. Notice that our current update rule requires expert information for every state, while a limited set of expert demonstrations is usually provided. Meanwhile, the indicator value in the expert correction may have conflict values if an expert provides multiple trajectories but chooses different actions at a state. Instead, our algorithm suggests storing expert data in the replay buffer and only performing expert correction when sampling expert data from the buffer.

Also, our algorithm weighs expert data to reduce off-policy biases. The importance sampling ratio is hard to calculate, so our algorithm uses a computationally-efficient weight

$$\left[ \frac{e^{\eta Q(s,a)}}{\sum_b e^{\eta Q(s,b)}} \right]^{\zeta}$$

where the power $\zeta$ is a rescaling factor to avoid miniature weights. This ratio assigns higher values to more frequent state-action pairs under the current $\epsilon$-greedy policy with respect to Q-estimates. The detailed algorithm is presented in algorithm 3 in the appendix.

## 6 Experiment

To evaluate the discernment of expert data, we test our algorithm using optimal and misleading expert data in a sparse-reward chain environment called DeepSea [Osband et al., 2019]. Then, we further test the data efficiency of our algorithm with the help of a limited amount of suboptimal human data on six complicated Atari games [Brockman et al., 2016]. We compare against other two discrete-action algorithms: Prioritized Dueling Double (PDD) Deep Q-learning Network (DQN) [Mnih et al., 2013, Wang et al., 2016, Hessel et al., 2018] with EZ-greedy [Dabney et al., 2020] exploration and Deep Q-learning from demonstrations (DQfD). The DeepSea environment requires less than one hour of training on an Nvidia 2070 GPU. Each Atari game requires 21 days of training on Compute Canada server with either one Nvidia V100 or one Nvidia P100.

## 6.1 Chain Environment

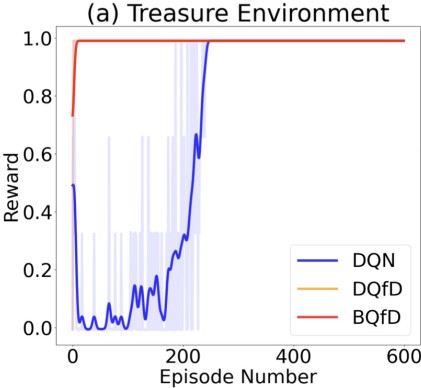 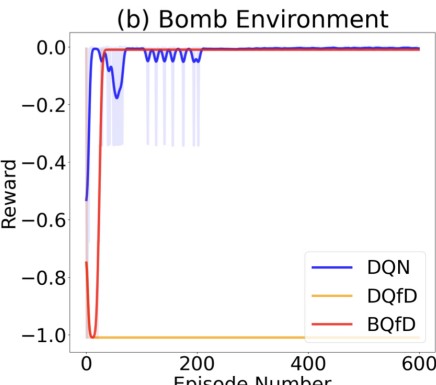

Figure 2: This figure shows agents' performances for the treasure case on the left and the bomb case on the right. It presents the training rewards versus the number of training episodes in an environment of size 50. Our algorithm can rapidly discover the optimal reward given the optimal expert data and acquire the optimal behavior given wrong expert information.

DeepSea is a deterministic chain environment of length $N$ with two actions: left and right. A figure to describe the environment is shown in the appendix. The agent is initialized on the left in each episode and terminates after $N$ steps. The agent receives rewards of $0$ for left actions and $\frac{-0.01}{N}$ for the right actions except for a fixed but unknown reward at the most right state. The unknown but fixed reward consists of either a treasure with a bonus of $+1$ or a bomb with a cost of $-1$. Regardless of the environment, we always provide agents with an expert trajectory towards the most right. Thus, in the treasure case, we test our algorithms' efficiency; in the bomb case, we evaluate our algorithms' robustness in the presence of highly misleading expert information. To provide a fair comparison, we include an expert trajectory to the replay buffer of DQN and provide no pretraining for any algorithms.

The results for the treasure case are shown on the left of figure 2. BQfD and DQfD can gain the optimal reward after one training episode much faster than DQN since the expert data directly leads to the reward, and these two algorithms extract expert information efficiently. But DQfD assumes the expert data to be optimal and causes catastrophic performance when expert demonstrations are misguiding, as seen in the bomb case, shown on the right of figure 2. On the contrary, our learning-from-demonstrations approach successfully learns the optimal policy, though the convergence speed is hindered due to confusing information. This result proves that BQfD efficiently extracts expert information and gradually stops using expert information.

## 6.2 Arcade Learning Environment

**Experiment Settings**

We further evaluated our approach on Arcade Learning Environments [Brockman et al., 2016, Mnih et al., 2013]: Breakout, ChopperCommand, Freeway, MsPacman, VideoPinball, and Seaquest. The architecture for all three algorithms is the same as PDD DQN [Mnih et al., 2013, Wang et al., 2016, Hester et al., 2018, Hessel et al., 2018] with prioritized replay buffer [Schaul et al., 2015]. Baselines' hyperparameters follow the choices in DQfD and prioritized replay buffer. Meanwhile, we tune hyperparameters for BQfD on the environment Seaquest, and the detailed results are shown in the appendix. All approaches are provided with one human-generated expert trajectory and are pretrained with this expert data. Given the randomness during training, all results shown are averaged over three random seeds.

| Game | Length | DQN | DQfD | BQfD |
|---|---|---|---|---|
| Breakout | 91M | $221 \pm 75$ | $291 \pm 119$ | $\mathbf{354 \pm 132}$ |
| ChopperCommand | 63M | $4493 \pm 1295$ | $7067 \pm 2403$ | $\mathbf{7377 \pm 2801}$ |
| Freeway | 23M | $\mathbf{30.4 \pm 0.4}$ | $28.2 \pm 0.8$ | $29.8 \pm 0.8$ |
| MsPacman | 76M | $3628 \pm 725$ | $\mathbf{4927 \pm 993}$ | $3550 \pm 613$ |
| Seaquest | 96M | $20595 \pm 8248$ | $22388 \pm 6294$ | $\mathbf{28546 \pm 11613}$ |
| VideoPinball | 116M | $8619 \pm 12582$ | $17235 \pm 15906$ | $\mathbf{30668 \pm 18772}$ |

Table 1: We present the average evaluation rewards over the previous 2M steps $\pm$ and the average standard deviations over the last 2M steps among three random seeds. BQfD shows apparent advantages in Breakout, ChopperCommand, Seaquest and VideoPinball and performs comparably to the best in Freeway and MsPacman.

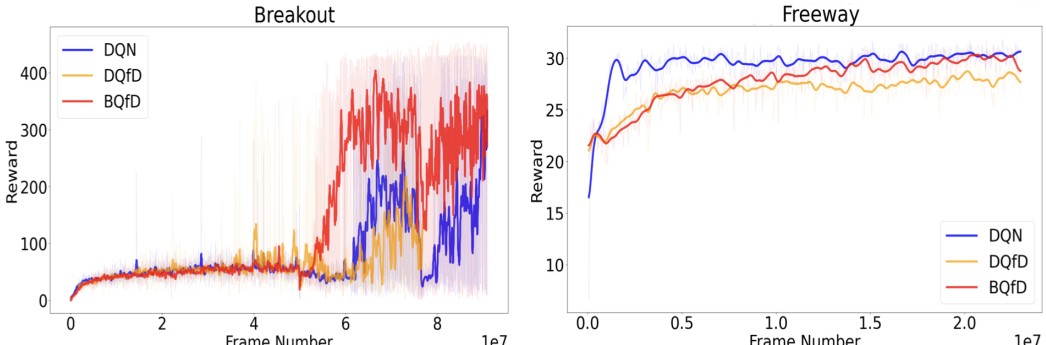

Figure 3: These figures present Breakout, ChopperCommand, SeaQuest, and Freeway's results averaged over three random seeds of BQfD, DQfD, and PDD DQN. We observe that BQfD's performance far exceeds other approaches and shows an obvious advantage in the late stage of learning.

**Results**

The overall results are shown in table 1. Here, we present final-learned evaluation rewards: the average means and standard deviations of evaluation rewards over the last 2M steps. Our algorithm outperforms the baselines in Breakout, ChopperCommand, Seaquest, and VideoPinball. In MsPacman, it still performs better than DQN but slightly worse than DQfD. In general, both three algorithms perform much worse than humans on MsPacman. DQN dominates the game in Freeway, but our algorithm recovers from weak expert data compared to DQfD.

The learning curve of Breakout and Freeway are presented in figure 3. Initially, the three algorithms' performances do not show significant differences. But the average evaluation performance slowly tilts in favor of BQfD; it learns fastest, and the order of DQN and DQfD varies. Surprisingly, BQfD's advantage arises in the latter stages of training instead of improving learning speed in the early stages. One reason might be that expert data leads to adequate explorations. Also, our approach encourages the agent to deviate from expert states and learn from trials and errors.

## 7 Discussion

Learning with demonstrations is an exciting direction for robotics tasks with continuous actions. Though our approach is limited to discrete action problems, it can be extended to continuous actions and policy gradient algorithms left as future work. Furthermore, a more accurate approximation of posterior variance should be considered for future work. Moreover, when experts are not limited to being optimal but instead generate informative trajectories, the quality of expert demonstrations becomes more challenging to judge. The task of selecting the correct expert becomes an important question.

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

# Appendix

## Derivation of Generalized Extended Kalman Filters

We derive the Q-value update rules under assumptions from the GEKF framework. The proof follows the steps in Sage and James (1997) [Sage and Melsa, 1971].

A Kalman filter estimates the hidden Q-values $Q_h$ of an agent at each discrete time step $t = 0, ..., T$ given noisy expert observations $A_{exp,h}$, $h = H - 1, ..., 0$ and noisy transition functions. The Linear Kalman Filter requires the previous state $Q_h$ and observation $A_{exp,h+1}$ to depend linearly on the current state $Q_{h+1}$. The extended Kalman filter relaxes these relationships to smooth functions. In this section, we discuss the generalized extended Kalman filter [Fahrmeir, 1992] where the observation $A_{exp,h+1}$ follows any exponential family distribution with the current state $Q_{h+1}$ as parameters. The probabilistic model is defined as:

$$q_H^* = 0, \tag{5}$$
$$q_h^* = \phi(q_{h+1}^*) + \nu_h, \quad \nu_h \sim N(0, \lambda I),$$
$$P(A_{exp,h}(s,a) = a | q_h^*) = exp(\eta q_h^*(s,a) - log(\sum_b exp(\eta q_h^*(s,b)))).$$

The linear and extended Kalman filters assume that the posterior is a Gaussian distribution; thus, they do not need to distinguish between conditional means and posterior modes. However, these two estimators are not equivalent to the generalized extended Kalman filter. This paper focuses on the maximum a posteriori estimator for a Kalman filter [Sage and Melsa, 1971]. Due to the Markovian assumptions of the model, the posterior distribution of hidden states is proportional to

$$P(q_{0:H}^* | A_{exp,0:H-1}, R_{0:H-1}) \propto \prod_{h=0}^{H-1} P(A_{exp,h} | q_h^*) \prod_{h=0}^{H-1} P(q_h^* | q_{h+1}^*, R_h) P(q_H^*),$$

Maximizing the posterior is equivalent to minimizing the negative log probability. By combining this objective with the probabilistic model in Equation 5, when $Q_H = 0$, we obtain:

$$- log P(q_{0:H}^* = Q_{0:H} | A_{exp,0:H-1}, R_{0:H-1})$$
$$\propto \frac{1}{2} \sum_{h=0}^{H-1} \|Q_h - \phi(Q_{h+1})\|_{(\lambda I)^{-1}}^2 - \sum_{t=0}^{H-1} \sum_{s,a} [\eta Q_h(s,a) - log(\sum_b exp(\eta Q_h(s,b)))],$$
$$=: \frac{1}{2} \sum_{h=0}^{H-1} \|Q_h - \phi(Q_{h+1})\|_{(\lambda I)^{-1}}^2 - \sum_{t=0}^{H-1} \psi(Q_h) \mathbb{1}[Q_H = 0],$$
$$=: J(Q_{0:H}).$$

Here, we write the log observation probability as $\psi(Q_h)$ and define the minimization objective $J(Q_{0:H})$, subject to equations in 5:

$$\min_{Q_{0:H-1}} J(Q_{0:H})$$
$$s.t. Q_H = 0,$$
$$Q_h = \phi(Q_{h+1}) + \nu_h, \ h = H - 1, ..., 0.$$

This optimization problem can be solved by exploiting Lagrangian multipliers $\alpha_0, ..., \alpha_H$. First, the equality constraints are combined into the loss to form a Lagrangian function:

$$L(Q_{0:H}, \lambda, \alpha_{0:H}) = \sum_{h=1}^{H} H(Q_h, \alpha_{h-1}, \nu_{h-1}) - \sum_{h=0}^{H} \alpha_h Q_h,$$

where

$$H(Q_h, \alpha_{h-1}, \nu_{h-1}) = \frac{1}{2} \|\nu_{h-1}\|_{(\lambda I)^{-1}}^2 - \psi(\phi(Q_h) + \nu_{h-1}) + \alpha_{h-1}(\phi(Q_h) + \nu_{h-1}).$$

Then, a minimal solution requires the gradients of the Lagrangian function over all variables to be zero at that point. This procedure results in a partial differential equation (PDE) system, also known as a two-point boundary value problem. For convenience, we denote the partial derivative of the log observation probability as:

$$l(Q_h) := \frac{\partial}{\partial Q_h}\psi(Q_h) = \frac{\partial}{\partial Q_h}\log(P(A_{exp,h}|q_h^* = Q_h)).$$

Then the PDE is presented as:

$$Q_h = \phi(Q_{h+1}) - \lambda[\frac{\partial\phi(Q_{h+1})}{\partial Q_{h+1}}]^{-1}\alpha_{h+1}, \tag{6}$$

$$\alpha_h = \left[\frac{\partial\phi(Q_{h+1})}{\partial Q_{h+1}}\right]^{-1}\alpha_{h+1} + l(Q_h),$$

$$\alpha_H = c, \quad \forall c \in \mathbb{R},$$

$$\alpha_0 = 0.$$

For simplicity, the differential equations above can be expressed as

$$Q_{h-1} = f(Q_h, \alpha_h, h), \tag{7}$$

$$\alpha_{h-1} = g(Q_h, \alpha_h, h).$$

This PDE system can be solved by the invariant embedding method. Our goal is to learn a function of $Q_{0:H-1}$ which only depends on constants and the time step $h$, denoted as

$$Q_h = r(\alpha_h, h) \text{ and } \alpha_h = C.$$

According to this function, Q-values at the previous step $h-1$ satisfy that

$$Q_{h-1} = r(\alpha_{h-1}, h-1).$$

By plugging in function $f$ and $g$ in equation 7, we obtain

$$r(g(r(C,h), C, h), h-1) = f(r(C,h), C, h). \tag{8}$$

Since those PDEs in equation 6 are all linear of Lagrangian multiplies $\alpha_{0:H}$, it is further assumed that the function $r(C, h)$ is a linear function of variable $C$. Thus $r(C, h)$ can be written linearly by $r(C, h) = D(h) - P(h)C$ where $D$ and $P$ are some smooth functions of step $h$. Since $r(C, h)$ must satisfy the boundary assumptions $Q_h = r(C, h)$ for any $C \in \mathbb{R}$, $D(h)$ must equal to $Q_h$ and thus $r(C, h) = Q_h - P(h)C$. After substituting it into equation 8, we obtain

$$Q_{h-1} - P(h-1)g(Q_h - P(h)C, C, h) = f(Q_h - P(h)C, C, h).$$

When deriving Q-value estimations backward, at each time step, we only maximize the posterior probability from this step until termination. Thus, we work on the boundary of the PDE with Lagrangian multiplier $C = \alpha = 0$. Therefore, the above equation holds at $C = 0$, and both sides' derivatives over variable $C$ are also equal. Thus, we gain the update rule:

$$Q_{h-1} - P(h-1)g(Q_h, 0, h) = f(Q_h, 0, h),$$

$$P(h-1)\frac{\partial g}{\partial C}(Q_h, 0, h) = -\frac{\partial f}{\partial C}(Q_h, 0, h).$$

The final step is to derive the initial conditions of $Q_H$, $P_H$. When the trajectory length is zero, we have a simple loss $|Q_H - 0|$, and obviously, the minimum point is $Q_H = 0$. Recall that the invariant embedding assumption gives $Q_H = r(\alpha_H, H) = Q_H - P(H)\alpha_H$ and the boundary value allows $\alpha_H$ to be any constant. Therefore, we conclude that $P(H) = \mathbf{0}$.

Now we have an exact per-step update rule given these initial values.

$$Q_{h-1} = \phi(Q_h) + P(h-1)\frac{\partial}{\partial Q_h}logP(A_{exp,h}|Q_h),$$

$$P(h-1) = \lambda I + [\frac{\partial\phi(Q_h)}{\partial Q_h}]^T P(h)\frac{\partial\phi(Q_h)}{\partial Q_h},$$

$$P(h-1) = [P^{-1}(h-1) - \frac{\partial^2}{(\partial Q_h)^2}logP(A_{exp,h}|Q_h)]^{-1}.$$

with initial conditions $Q_H = 0$ and $P(H) = \mathbf{0}$. This provides the final Q-values update rule as seen in algorithm 1.

---
**Algorithm 2** BQfD
---
**Input:** Expert actions $A_{exp}(s)$ for all states, a constant $\beta$.
**Initialize:** Q-value functions $Q_{0:H}^0 = 0$.
**for** Each episode l=1,2,... **do**
  **Sample:** Initialize $S_0^l$ and sample a trajectory of length H by choosing $A_h^l$ with maximum Q-value. Observe $\{R_h^l\}_{h=0}^{H-1}$.
  **for** Each time step h=H-1,...,0 **do**
    **Q-value update**
    $$Q_h^l(s,a) = \begin{cases} \alpha_n[R_h^l + \gamma max_a Q_{h+1}^l(s_{h+1}, a)] + (1-\alpha_n)Q_h^{l-1}(s_h, a_h), & if\ (s,a) = (s_h^l, a_h^l) \\ Q_h^{l-1}(s,a), & otherwise \end{cases}$$
    **if** $s_h$ is visited in expert demonstration **then**
      **Compute Weight Decay**
      $w_n = \frac{\beta^2 + 4n^l(s_h, a_h)}{(\beta + n^l(s_h, a_h))^2}$
      **Expert Correction**
      $Q_h^l(s_h, a_h) += \eta w_n (\mathbb{1}[a_h = A_{exp}(s_h)] - \frac{exp(\eta Q_h^l(s_h, a_h))}{\sum_{b \in \mathcal{A}} exp(\eta Q_h^l(s_h, b))})$
    **end if**
  **end for**
  **Initialize:** Q-value functions for next episode $Q_{H+1}^{l+1} = Q_0^l$
**end for**
---

## Algorithm

Based on the GEKF model, our estimations can be computed step-by-step as presented in algorithm 1 in a backward view. The per-step update rule comprises a prediction step, the sample-based Bellman update and a correction step. Note that $W$ is the posterior covariance term in Kalman filters.

Our BQfD algorithm shown in algorithm 2 is to avoid the matrix inversion and Hessian computation in the update of weights $W$. We only update the Q-value of the current state-action pair $(s_h, a_h)$ and approximate the weight of $\{(s_h, a_h), (s_h, a_h)\}$ by a $O(\frac{1}{n^l(s_h, a_h)})$ term where $n^l(s_h, a_h)$ is the visitation time of this state-action pair until the episode $l$.

Next, we extend our algorithm and include deep neural networks. We use TD loss for non-expert transitions denoted as $J(theta)$ and use en expert-information embedded loss $J_{exp}(\theta)$ for expert transition, defined as:

$$J_{exp}(\theta_{main}) = (R(s,a) + \gamma Q(s', a^{max}; \theta_{target})$$
$$+ w_n \eta(1 - \frac{exp(\eta Q(s_h, a_h; \theta_{main}))}{\sum_{b \in \mathcal{A}} exp(\eta Q(s_h, b; \theta_{main}))} - Q(s, a; \theta_{main}))^2$$

where $\theta_{target}$ is the parameters of target network, $\theta_{main}$ are the parameters of the main network, and $a^{max} = argmax_a \tilde{Q}(s', a; \theta_{target})$. We also leverage prioritized replay buffer [Schaul et al., 2015] to balance the amount of expert data and online training data. The detailed algorithm is presented in algorithm 3.

## Experiment Setting

The DeepSea environment requires less than one hour of training on an Nvidia 2070 GPU. Each Atari game requires 21 days of training on Compute Canada server with only 1 GPU and 24GB memory. The GPU on Compute Canada is either Nvidia V100 or Nvidia P100. All codes are provided in the supplementary material.

### DeepSea

As shown in figure 4, DeepSea is a deterministic chain environment represented as a square grid of size $N \times N$. All agents are initialized at the top-left corner. At each time step, the agent can move left or right but always moves down by one. Notice that the edges of the grid bound the agent's

---

**Algorithm 3** BQfD deep

---

**Initialize:** $D^{replay}$: initialized with demonstration data set, $\theta_{main}$: random weights for main network (random), $\theta_{target}$: random weights for target network (random), $\tau$: target network update frequency

**Pretrain:** Randomly sample from replay buffer and minimize loss $J_{exp}(\theta)$

**for** step t=1,2,... **do**

    Take action $a$ at state $s$ with maximum current Q-values $\tilde{Q}(s, a; \theta_{main})$

    Play action $a$ and observe $(s', r)$

    Store $(s, a, r, s', terminal)$ into prioritized replay buffer $D^{replay}$

    Sample a mini-batch from $D^{replay}$ according to prioritization

    **if** data is from expert **then**

        Minimize $J_{exp}(\theta_{main})$ and perform a gradient descent step to $\theta_{main}$

    **end if**

    **if** data is newly drawn **then**

        Minimize $J(\theta_{main})$ and perform a gradient descent step to $\theta_{main}$

    **end if**

    **if** t mod $\tau$ =0 **then**

        Update $\theta_{target}$ by $\theta_{main}$

    **end if**

**end for**

---

movements. Each game terminates whenever the agent reaches the bottom. The agent receives rewards of 0 for left actions and $\frac{-0.01}{N}$ for the right actions except for a fixed but unknown reward at the bottom-right state. The unknown reward consists of either a treasure with a bonus of $+1$ or a bomb with a cost of $-1$. DeepSea suffers from both sparse rewards and small negative rewards for movements towards the unknown reward. Thus, it is a suitable environment to test the efficiency of guided exploration.

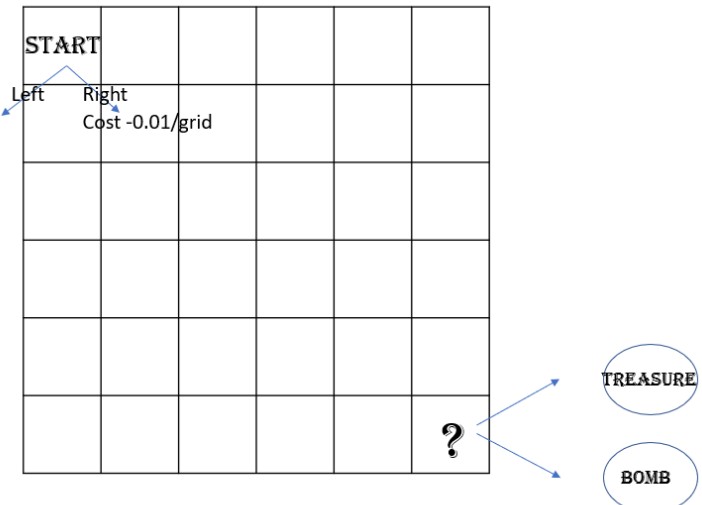

Figure 4: The chain environment consists of a grid and two actions, left and right. At each step, the agent always moves down by one. The agent receives a fixed but unknown reward at the bottom-right corner, either a treasure or a bomb. The game ends whenever the agent reaches the bottom.

The architecture consists of two layers of Perceptron with Relu activation functions. It is followed by two dense layers with 256 hidden units that output value function and advantage function. The

| BQfD Hyperparameter | |
| --- | --- |
| learning rate | 0.0005 |
| eta | 3 |
| variance | 0.9 |
| **DQN Hyperparameter** | |
| learning rate | 0.05 |
| exploration decay start frame | 300 |
| **Bootstrapped DQN Hyperparameter** | |
| learning rate | 0.05 |
| number of ensembles | 20 |
| **DQfD Hyperparameter** | |
| learning rate | 0.05 |

Table 2: We present the values of hyperparameters for all algorithms tuned on a game of size 20.

| BQfD Hyperparameter | |
| --- | --- |
| learning rate | 0.000055 |
| variance | 22 |
| expert decay ratio | 4 |
| **DQN or DQfD Hyperparameter** | |
| learning rate | 0.0000625 |

Table 3: We present the values of hyperparameters in the table. These hyperparameters for BQfD are tuned on Seaquest and used for all environments except MsPacman; because all three algorithms do not perform well on MsPacman, and its hyperparameters are specifically tuned. The hyperparameters for DQN and DQfD are the same as in the original papers.

state is the current position denote by a 2-dimensional vector. The hyperparameters of BQfD are set as follows: learning rate $lr = 0.0005$, variance $\lambda = 0.6$, and expert policy parameter $\eta = 3$. The learning rate for other methods is tuned to be $lr = 0.05$. Detailed tuning results on game of grid size $N = 20$ are shown in table 2.

**Atari**

We further evaluated our approach on Arcade Learning Environments: Breakout, ChopperCommand, Freeway, MsPacman, VideoPinball, and Seaquest. The architecture for all three algorithms is the same as PDD DQN; it consists of three layers of convolution neural networks of filter sizes 32, 64, and 64, kernel sizes [8, 8], [4, 4], and [3, 3], and strides of 4, 2, 1 respectively where each layer has a ReLU activation function. Furthermore, the output is passed to two dense layers, which output the value and advantage functions. As in DQfD, the input features are grey-scaled Atari frames resized to [84, 84] with a history length of 4. Furthermore, we scale down the reward during training to allow for standardized learning rates for all games, and each reward $r$ is scaled as $sign(r)log(1 + r)$.

We reuse hyperparameters from DQfD and prioritized replay buffer for the two baselines. For BQfD, we tune our hyperparameters on the environment Seaquest, including learning rate, variance, and expert priority decay ratio. However, all three algorithms cannot learn the game MsPacman. So we tune the learning rate and initial random exploration ratio on MsPacman separately. The hyperparameters are chosen as shown in table 3: learning rate $lr = 0.000055$ and variance $\lambda = 22$. The expert priority decay speed is set to 4, meaning priorities of expert data decay until $4 \times 30$ millions of simulation steps. The initial epsilon greedy exploration ratio is 1.0 for MsPacman and 0.0 for others. All approaches are provided with one human-generated expert trajectory and are pretrained with this expert data. The expert trajectory is generated by a human player who understands the game's rules but occasionally makes mistakes. For Breakout, ChopperCommand, Freeway, MsPacman, Seaquest, and VideoPinball, the expert demonstrations contain a single trajectory of 110, 33800, 27, 30k, 34k, and 82k rewards, respectively. Furthermore, we ensured that, in total, the expert demonstrations consist of around 6k frames; note for Freeway, we repeated the same trajectory thrice as its maximum length of a single game is 2049.

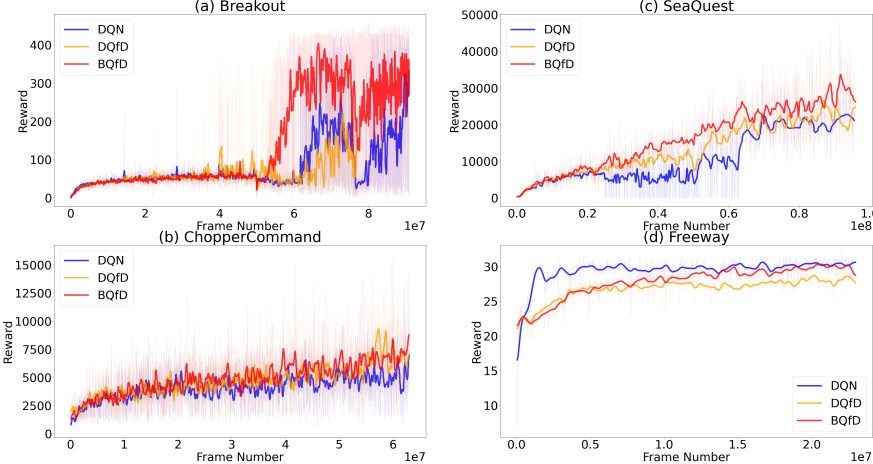

Figure 5: These figures present Breakout, ChopperCommand, SeaQuest, and Freeway's results averaged over three random seeds of BQfD, DQfD, and PDD DQN. We observe that BQfD's performance far exceeds other approaches and shows an obvious advantage in the late stage of learning.

The training on Atari games requires reloading due to time limits, but the replay buffer is not saved. The counting of visitation times for BQfD is reset to mean previous visitation times among expert data. Thus, there may exist a performance downgrade in the middle.

