# OpenReview forum: "Bayesian Q-learning With Imperfect Expert Demonstrations"
_NeurIPS.cc/2022/Workshop/Offline_RL — Offline RL Workshop NeurIPS 2022_

### Official Review · Reviewer_AA7Z · 2022-10-18
**Paper review**

**Rating:** 7
**Confidence:** 3

**Review:**

### Summary

In RL, it is common to supplement on-policy exploration with (off-policy) expert demonstrations. In practice, the so-called "experts" can be a bit noisy, so their demonstrations aren't necessarily optimal. This paper proposes a Bayesian approach that accounts for suboptimal experts. In short, the expert is modeled as a Boltzmann distribution (i.e., softmax) on the optimal Q-values, so that the sampled action is not necessarily optimal. Formulated as such, the learning agent maximizes the posterior probability of the Q-values given the expert demonstrations. The posterior is defined as a generalized extended Kalman filter (GEKF) -- basically, a Kalman filter but with nonlinear mean functions (typically, neural nets) -- thereby supporting Q-value calculation via a forward-backward algorithm (if I've understood correctly). Empirically, the proposed method, BQfD, seems to rely less on noisy expert demonstrations than its non-Bayesian counterpart, DQfD. Further experiments with the Arcade Learning Environment show that BQfD performs either better, or not much worse, than the baselines.

### Strengths

* It's a nice, principled extension of DQfD A Bayesian approach makes total sense.
* BQfD seems to perform very well relative to the baselines.

### Weaknesses

* Sections 4.3 & 4.4, which derive and discuss the algorithm, were a bit hard for me to follow. I think this is just a matter of not having enough space to go into detail or provide intuition. (Admittedly, I did not bother to read the appendix on the derivation.) In particular, it's not really clear to me how the proposed approach automatically adjusts its reliance on the (possibly noisy) expert demonstrations. I'm guessing it's all in the posterior update, which, like Bayesian linear regression (or linear Thompson sampling), probably accumulates the online exploration in the covariance matrix. The paper could make this a bit clearer, since robustness to noisy experts is its main claim.
* The DeepSea environment seems a bit trivial to me. The problem essentially boils down to "go or don't go"; no need to make it a sequential decision process. This deep-dive (pun intended) would have been more convincing with a more interesting environment.

### Other comments/questions

* I wonder if anything can be said about the optimality or convergence rate of BQfD. Does it simply inherit existing theory for Q-learning, or can you show that it improves on known rates?

---

### Official Review · Reviewer_jQkd · 2022-10-20
**An interesting paper with limited experimental results.**

**Rating:** 7
**Confidence:** 3

**Review:**

This paper proposes a Q-value estimation under the generalized extended Kalman 105 filter (GEKF) and proposes the Bayesian Q-learning from Demonstrations (BQfD) to implement such an update rule. The BQfD indeed demonstrates superior performance among some Atari games. The reviewer notices that the number of seeds is small (3 seeds) and the variance of Breakout, ChopperCommand, MsPacman, Seaquest, VideoPinball are too large. While the proposed method indeed shows superior performance in VideoPinball, it is hard to tell whether the proposed method indeed improves the performance in Breakout, ChopperCommand, MsPacman, and Seaquest.

Nevertheless, the proposed method is interesting and the reviewer believes it can be further implemented to achieve better performance.